# Incidence and Risk Factors of Alcohol Relapse after Liver Transplantation: Analysis of Pre-Transplant Abstinence and Psychosocial Features

**DOI:** 10.3390/jcm9113716

**Published:** 2020-11-19

**Authors:** Tien-Wei Yu, Yu-Ming Chen, Chih-Chi Wang, Chih-Che Lin, Kuang-Tzu Huang, Yueh-Wei Liu, Li-Wen Hsu, Wei-Feng Li, Yi-Chai Chan, Chao-Long Chen, Chien-Chih Chen

**Affiliations:** 1Department of Psychiatry, Kaohsiung Chang Gung Memorial Hospital, Chang Gung University College of Medicine, Kaohsiung 833, Taiwan; mp9596@cgmh.org.tw (T.-W.Y.); yuming0320@cgmh.org.tw (Y.-M.C.); 2Liver Transplantation Center and Department of Surgery, Kaohsiung Chang Gung Memorial Hospital, Chang Gung University College of Medicine, Kaohsiung 833, Taiwan; ufel4996@ms26.hinet.net (C.-C.W.); chihchelin@cgmh.org.tw (C.-C.L.); anthony0612@cgmh.org.tw (Y.-W.L.); hsuliwen@ms55.hinet.net (L.-W.H.); webphone@cgmh.org.tw (W.-F.L.); sparkle@cgmh.org.tw (Y.-C.C.); 3Institute for Translational Research in Biomedicine, Kaohsiung Chang Gung Memorial Hospital, Chang Gung University College of Medicine, Kaohsiung 833, Taiwan; huangkt@cgmh.org.tw

**Keywords:** alcohol-associated liver disease, liver transplantation, abstinence, alcohol, relapse, psychosocial features

## Abstract

Alcohol-associated liver disease (ALD) is a common indication for liver transplantation (LT). Alcohol relapse after LT is associated with graft loss and worse prognosis. Over the past 20 years, the number and prevalence of living donor liver transplantations (LDLTs) have increased in Taiwan. The aims of this retrospective study are to analyze the incidence and risk factors of alcohol relapse after LT at a single center in Taiwan. A total of 98 patients with ALD who underwent LT from January 2012 to December 2018 were retrospectively evaluated by chart review. Pre-transplant characteristics as well as psychosocial and alcoholic history were used to test the possible associations among the risk factors studied and post-LT alcohol relapse. The incidence of post-LT alcohol relapse was 16.3%. The median duration of alcohol relapse after liver transplantation was 28.1 months (range: 1–89.4 months). The cumulative incidence was 12% and 19% at 1 year and 3 years after LT, respectively. The most powerful risk factors were a pre-LT abstinence period less than 6 months and younger age of starting alcohol. For predicting alcohol relapse, the accuracy rate of abstinence less than 6 months was up to 83.7%. In summary, pre-abstinence period plays a role in predicting post-LT alcohol relapse. Post-LT interventions should be considered specifically for the patients with short abstinence period. Long-term follow-up, patient-centered counseling, and enhancement of healthy lifestyle are suggested to prevent alcohol relapse.

## 1. Introduction

Alcohol-associated liver disease (ALD) is the most common cause of cirrhosis in Western countries [1,2]. In the United States, ALD has accounted for an increasing percentage of liver transplantation (LT) [1,2]. LT evaluation for patients with alcohol use disorder (AUD) focuses on risk of post-LT alcohol relapse. Depending on the definition of relapse, the risk of alcohol relapse ranges from 8% to 22% at 1 year, 25% to 35% at 3 years, and up to 40% at 5 years after LT [3,4,5,6,7]. Alcohol relapse after LT is associated with reduced overall survival because the graft might be damaged from direct toxicity of alcohol or from poor drug adherence with the immunosuppressive regimen. Several studies have reported that LT patients who relapse have worse outcomes in 5-year and long-term follow-ups [8,9,10,11]. In contrast, Lombardo-Quezada et al. demonstrated that although alcohol relapse after LT is associated with increased risk of graft cirrhosis, the patients’ survival rate showed no significant difference compared with non-relapse patients [12].

Despite the alcohol relapse rate in approximately 20–25% of ALD patients post-transplantation [8], the ALD patients with no relapse post-transplantation present no significant difference in prognosis and clinical course in comparison to patients with other etiologies of liver disease [13,14,15]. As a consequence, the investigation of risk factors of alcohol relapse after liver transplantation has drawn a great deal of attention. In contrast to non-ALD patients, ALD patients have an increased risk of psychological and psychosocial impairment [16]. Recently, investigators have examined the risk factors of alcohol relapse, including age, sex, social support, employment, cigarette smoking, pre-transplant psychiatric condition, and abstinence period [11,17,18,19,20]. The prevalence of psychiatric disorders, predominantly depressive and anxiety disorders, was approximately 65% in patients with ALD [3,21,22]. Psychiatric disorders were associated with a greater risk of post-LT alcohol relapse [3,17,23,24]. DiMartini et al. [3] reported that a history of pre-transplant depressive disorder predicted the time to first alcohol relapse. Availability of family support was also a protective factor of alcohol relapse [11,20]. A careful assessment of the risk factors and timely enhancement of psychosocial support should be performed to optimize the care before and after liver transplantation.

In Taiwan, the number and prevalence of living donor liver transplantations (LDLTs) have increased in the past 20 years [25,26]. According to the increase in hospitalized ALD patients and accumulative LT cases [27], our study aims to analyze the incidence and risk factors of post-LT alcohol relapse.

## 2. Study Design

### 2.1. Study Participants

We included all the patients diagnosed with ALD who underwent LT from January 2012 to December 2018 at Kaohsiung Chang Gung Memorial Hospital retrospectively. At the time of the medical records review, these patients were at least 6 months post-transplant and discharged from the hospital. A total of 898 patients underwent LT, and 106 patients (11.8%) with ALD were identified. Four died within one and a half months after the transplant, and four had incomplete pre-LT psychosocial data in recording and were not included in the study (Figure 1).

Before transplantation, complete evaluation is performed by hepatologists and transplant psychiatrists. Experienced transplant psychiatrists interviewed patients with AUD as part of routine psychosomatic clinical care. AUD is categorized as alcohol abuse and alcohol dependence based on the Diagnostic and Statistical Manual of Mental Health Disorders, Fourth Edition, Text Revision (DSM-IV-TR) criteria [28]. The data collection had adequate provisions in place to protect the confidentiality of the data. The patients’ characteristics included age, gender, donor–recipient relationships, hepatitis status, marital status, occupational status, education level, cigarette smoking status, age at drinking onset, alcohol remission status, family history of alcoholism, pre-transplant alcohol abstinence period (months without alcohol consumption; the data were recorded as a quantitative and continuous variable), Hamilton depression rating scale (HAM-D), and previous and current mental illness other than alcoholism. Family function was measured using the Chinese version of the family APGAR (Adaptability, Partnership, Growth, Affection, and Resolve) index as in our previous study [29]. The pre-LT alcohol consumption level was evaluated using the high-risk alcoholism relapse (HRAR) scale [30]. Three items, including duration of heavy drinking (years), daily number of drinks, and number of prior inpatient alcoholism treatments are included in the scale. A score of 0–3 indicates low risk and 4–6 indicates high risk [30,31].

After transplantation, patients had follow-up visits at different departments of outpatient clinics in Kaohsiung Chang Gung Memorial Hospital. These outpatient clinics include hepatology, nephrology, cardiology, and otorhinolaryngology. During the follow-up visits, all the patients were asked about alcohol consumption and time of first alcohol use after LT. Alcohol relapse was documented in the outpatient medical records and exclusive of social and occasional alcohol use defined as sustained alcohol use. Assessment of alcohol relapse was based on patients’ statements and additional information by family members. Our primary outcome variable was the incidence of post-LT alcohol relapse.

All patients in the study were Taiwanese citizens. The Ethics Committee of CGMH, Taiwan (Approval No CGMH 201900889B0) waived the need for patient consent.

### 2.2. Statistics

Univariate and multivariate binary logistic regressions were performed to evaluate the association between patient characteristics and post-LT alcohol relapse and to identify possible risk factors. Statistical significance was defined as alpha = 0.05. Categorical variables are expressed as proportions, and continuous variables are expressed as the mean ± standard deviation (SD). Multivariate forward logistic regression analysis was conducted to identify significant predictors of alcohol relapse. Receiver operating characteristic (ROC) curve analysis was evaluated the cut-off point of pre-LT alcohol abstinence duration. Post-LT alcohol use rate was estimated using Kaplan–Meier method, with follow-up time starting at LT date and ending at date of first alcohol use. The Statistical Package for the Social Sciences (SPSS Version 22 for Windows, SPSS Inc., Chicago, IL, USA) was used for all analyses.

## 3. Results

### 3.1. Demographic and Clinical Characteristics in LT Recipients

The 98 subjects’ demographics and characteristics are presented in Table 1. The mean age was 51.5 ± 7.0 years old, and 94.9% patients were male. The majority of donations were LDLTs (94.9%). Approximately 90% of the patients were married and smoked cigarettes. The highest achieved educational level of the most patients was senior high school (46.9%). Among the 98 patients, 76 (77.6%) were retired, 11 (11.2%) were employed. Forty six patients (46.9%) underwent LT due to alcohol-related liver disease, and the other 52 patients (53.1%) had hepatocellular carcinoma (HCC) plus alcohol-related liver disease. The Model for End-stage Liver Disease (MELD) score presented a mean value of 16.4 ± 6.9. Most patients started to drink in early adulthood at the age of 20.1 ± 4.4 years. Concerning the pre-LT abstinence, 30 patients (30.6%) were abstinent for ≤6 months, and 68 patients (69.4%) were free from alcohol >6 months, with a mean duration of 20.7 ± 27.9 months. A family history of alcohol consumption among immediate family and among siblings was reported in 30 (30.6%) and 20 (20.4%) patients, respectively. Regarding alcohol use disorder, 8 patients met the criteria for alcohol abuse and 90 patients for alcohol dependence.

### 3.2. Predictors of Alcohol Relapse

In this study, the median of patients’ follow-up time after LT was 42.7 months (range: 6.2–89.4 months). All post-LT patients were divided into an alcohol relapse group and a non-relapse group. Comparisons of the post-LT alcohol relapse and non-relapse groups are presented in Table 2.

Univariate analysis further revealed that the relapse group was more likely to have been abstinent from alcohol for less than 6 months before transplant (36.7% versus 7.4% in the non-relapse group; *p* = 0.001). Moreover, the alcohol relapse group had a younger mean age of starting alcohol (17.4 years old versus 20.6 years; *p* = 0.010). The other risk factors did not achieve statistical significance (all *p* > 0.05). In addition, HRAR is a scoring system to predict the risk of alcohol relapse after LT. Among the 98 patients, 68 (69%) had complete HRAR scores. Statistical analysis revealed that high risk scores were not associated with high relapse rate post-LT (*p* = 0.895). Multivariate logistic regression analysis based on the independent predictor variables indicated that younger age of starting alcohol and abstinence for less than 6 months were significant at the 0.05 probability level (Table 3).

### 3.3. ROC Curve of Alcohol Abstinence Durations for Alcohol Relapse

The area under the ROC curve (AUC) for the combination of alcohol relapse and non-relapse in comparison with durations of alcohol abstinence before LT was 0.753 (Figure 2A). Kaplan–Meier estimates for patients’ alcohol relapse rate after LT are shown in Figure 2B. The median duration of alcohol relapse after liver transplantation was 28.1 months (range: 1–89.4 months). The cumulative incidence of any alcohol use was 12% at 1 year and 16% at 2 years after LT. For long-term follow-up, the cumulative probability was 19% and 23% at 3 and 5 years after LT, respectively.

The sensitivity and specificity of alcohol abstinence for alcohol relapse at the optimal cut-off of 5.7 months were 0.793 and 0.688, respectively. Moreover, the positive predictive value and negative predictive value for less than 6 months abstinence prior to LT were 68.8% and 76.8%, respectively (Table 4). For predicting alcohol relapse, the accuracy rate of abstinence less than 6 months was 83.7% of the patients.

## 4. Discussion

In Asia, the number of patients with ALD who have undergone LT has been increasing in recent years [32]. In our center, LT for ALD also gradually increased from <5% in 2008 to approximately 21% in 2018. AUD is an addictive disorder that includes compulsive behaviors and relapses as core symptoms [33]. Alcohol relapse after LT brings negative impacts, including graft dysfunction, reduced survival, and negative public perception of LT [34]. In addition, the prediction of alcohol relapse requires evaluation and is also one of the cost-effectiveness strategies for relapse prevention. Other cost-effectiveness strategies for relapse prevention include psychosocial intervention, such as motivational enhancement therapy (MET), cognitive behavioral therapy (CBT), and twelve-step facilitation (TSF) [35]. In the present study, we evaluated the demographic and psychosocial factors associated with alcohol relapse. Our study found that the pre-transplant abstinence period was a strong predictor of alcohol relapse. The patients with ALD and less than 6 months of abstinence before LT had higher risk of post-LT alcohol relapse, and close post-transplant follow-up is suggested. Long-term follow-up, patient-centered counseling, and enhancement of healthy lifestyle are suggested to prevent alcohol relapse.

In addition to the pre-transplant abstinence period, our data indicate that a younger mean age of starting alcohol is also a predictor of post-LT alcohol relapse. This result is analogous to a previous study which elucidates the association between early drinking and later problems of alcohol abuse, even when other risk factors are controlled [36]. Recent evidence suggests that transplant patients who relapsed to alcohol consumption have a younger median age [7,10,11,17,37]. It is known that young adults are at higher risk of excessive alcohol use than other age groups [38]. Furthermore, the young adults with alcohol use problems tend to drink in a more dangerous pattern, such as binge-drinking, in order to achieve the drunk status more rapidly [39]. Younger patients need to be under stringent follow up during the post-transplantation episode to prevent relapse. Comprehensive medical care and alcohol rehabilitation programs should be established to help these patients. In multivariate regression analysis, the age of starting alcohol use had less of an impact on relapse than abstinence < 6 months. A possible explanation is that, in comparison with the age of starting alcohol use, the patients with abstinence < 6 months may suggest poor self-control in alcohol use, which enhances the risk of alcohol relapse post-LT. A recent study reported that the ALD patients with greater trait self-control were more likely in the high-functioning group at three years following treatment [40]. Another study indicated that the ALD patients with low internal locus of control in programs with low external control were more probable to use alcohol during residential treatment [41].

Satapathy et al. showed that the patient support system was a significant predictor of alcohol relapse [11]. Support from one’s immediate family, such as from a spouse, parent, or child, could protect against alcohol relapse and help patients recover after LT [11]. The present study also drew comparisons among subgroups based on donation type and family support. Unexpectedly, the donation type and family support did not significantly affect the occurrence of post-LT alcohol relapse in our study. One possible reason for this discrepancy is that approximately 96% of the patients underwent LDLT, and almost all donation types were first-degree relatives of the recipients (approximately 70%) in this study. Our previous research has shown that for the family functional score, 76% of patients feel “Satisfaction” [29]. Similarly, recipients showed greater satisfaction with family function in this study. However, the family support system still plays an important role in predicting alcohol relapse after LT [11,42]. For non-immediate family and family dysfunction, attention should be paid to this group to prevent alcohol relapse.

In summary, recipients with shorter pre-LT alcohol abstinence periods have a greater tendency of relapse, which is related to adverse outcomes after transplant with psychological stress. We suggest that these patients should be involved in intensive post-transplant care to prevent possible alcohol relapse and related negative physical influence.

There are several strengths of our study design. In the last 20 years, LDLT accounted for over 90% of liver transplants in Asia [43]. We have the largest sample size of LDLT in a single-center study. By the end of 2018, 1587 of the 1882 (84.3%) liver transplants performed at our center were LDLT [43]. The psychosomatic status of the patient was evaluated by the same clinical psychiatric doctor, and the data were collected as part of the routine psychosomatic clinical care that was not influenced by other personal involvement. Moreover, multiple factors affect the prognosis after LT, including disease diagnosis, advancement of surgical techniques, clinical care, etc. The strength of our current study is that all treatments were performed in a single center, so potential influences such as surgical procedures and patient care were minimized. We adopted the method of involving multidisciplinary outpatient clinics to detect alcohol relapse. Self-reporting first alcohol relapse during follow-ups is somewhat stressful and embarrassing. Having a good rapport with physicians at different departments of outpatient clinics increases the chances of reporting truthfully.

The limitations of this study are those inherited from retrospective and single-center studies. The single-center criteria of LDLT may introduce selection bias into this study. However, single-center analyses have advantages. Pretransplant evaluation is performed by the same group of transplant team members. After transplantation, most LT recipients rent the suggested houses nearby our hospital for months so that they can visit follow-up outpatient clinics as scheduled. A retrospective review of medical records is more likely to underestimate the true prevalence of post-LT alcohol relapse. We trace the medical recordings from multidisciplinary outpatient clinics to decrease the risk of underreporting. Without unified notification in advance, the monitoring and recording of alcohol relapse at every clinic depends on the physician’s individual habit. By reviewing their medical records, we can detect alcohol relapse for the first time, but the data are not adequate for quantitative analysis. Prospective studies with an emphasis on the standard assessment of alcohol relapse by physicians may strengthen our conclusions. An important limitation should be mentioned. In our study, 88.8% of patients were smokers. The smoking behavior is associated with post-LT alcohol relapse, according to a recent systematic review [24]. However, we lacked detailed surveys of the smoking behavior, such as the Fagestöm test. Additionally, previous studies concerning correlation between alcohol relapse and the post-LT survival revealed inconsistent results. In this study, due to relatively short follow-up time and few deceased patients, it is inappropriate to investigate the relationship between alcohol relapse and survival. Further advanced research is needed to elucidate possible correlations.

## 5. Conclusions

Our study indicates that in the patients receiving LDLT in Taiwan, the pre-transplant abstinence period is a powerful predictor for post-LT alcohol relapse. For patients with a short pre-transplant abstinence period, additional well-designed interventions after LT are strongly recommended. Long-term follow-up, patient-centered counseling, and enhancement of a healthy lifestyle are suggested.

## Figures and Tables

**Figure 1 jcm-09-03716-f001:**
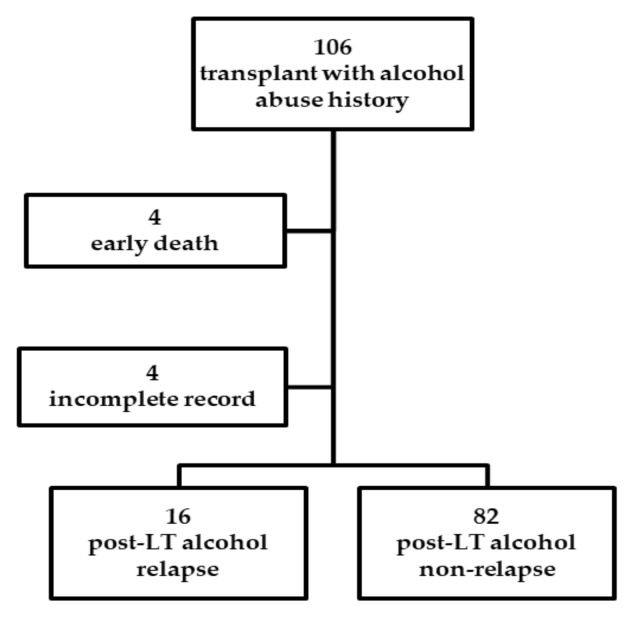
Patient enrollment and flowchart of the study results.

**Figure 2 jcm-09-03716-f002:**
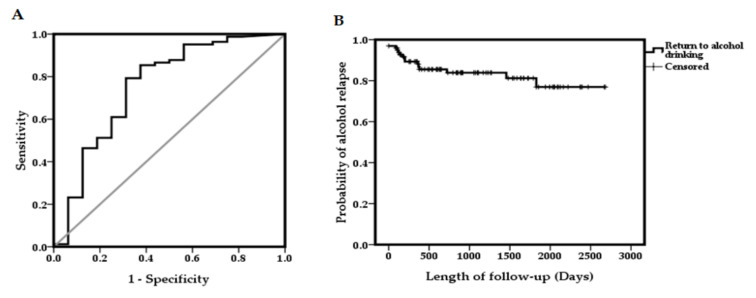
(**A**) ROC curve for abstinence during predictive alcohol relapse. (**B**) Cumulative survival to time of first alcohol use after LT.

**Table 1 jcm-09-03716-t001:** Demographic characteristics of the inclusive patients.

Variable	Total (n = 98, n%)	Variable	Total (n = 98, n%)
**Gender (n %)**		**Diagnosis**	
Male	93 (94.9)	Alcoholic cirrhosis	46 (46.9)
Female	5 (5.1)	Alcohol + HCC ^1^	52 (53.1)
**Age (y/o, mean ± SD (range))**	**51.5 ± 7.0 (36–68)**	**Alcohol start age (y/o, mean ± SD (range))**	**20.1 ± 4.4 (12–30)**
Male	51.8 ± 6.9 (37–68)	**Pre-LT abstinence (months, mean ± SD(range))** **^2^**	20.7 ± 27.9 (0–122.2)
Female	47.0 ± 6.5 (36–53)	**Pre-LT abstinence duration**	
**MELD score** **^3^**	16.4 ± 6.9 (6–36)	>6 months	68 (69.4)
**Donation type**		<6 months	30 (30.6)
Children (son/daughter)	39 (39.8)/30 (30.6)	**HAMD score (mean ± SD (range))** **^4^**	3.6 ± 2.7 (0–13)
Siblings	10 (10.2)	**Family APGAR** **index (mean ± SD (range))** **^5^**	8.8 ± 1.5 (3–10)
Spouse	8 (8.2)	**Family history of alcoholism**	
Other relatives	6 (6.1)	None	48 (49.0)
Cadaveric donor	5 (5.1)	Immediate family	30 (30.6)
**Marital status**		Siblings	20 (20.4)
Single	3 (3.1)	**Previous diagnosis of mental illness**	
Married	87 (88.8)	No	86 (87.8)
Separated/divorced	8 (8.2)	Yes	12 (12.2)
**Educational level**		**Current diagnosis of mental illness**	
Primary school	16 (16.3)	No	81 (82.7)
Junior high school	27 (27.6)	Yes	17 (17.3)
Senior high school	46 (46.9)	**Alcohol use disorder**	
College or above	9 (9.2)	Alcohol abuse	8 (8.2)
**Occupation**		Alcohol dependence	90 (91.8)
Housewife/Unemployed	11 (11.2)	**Alcohol relapse**	
Retired	76 (77.6)	No	82 (83.7)
Employed	11 (11.2)	Yes	16 (16.3)
**Smoke**			
No	11 (11.2)		
Yes	87 (88.8)		

Data are presented as mean ± SD unless otherwise specified. ^1^ HCC: hepatic cellular carcinoma; ^2^ LT: liver transplantation; ^3^ MELD: Model for End-stage Liver Disease; ^4^ HAMD: Hamilton Depression Rating Scale; ^5^ APGAR–Adaptation, Partnership, Growth, Affection, Resolve.

**Table 2 jcm-09-03716-t002:** Univariate analyses of pre-transplant risk factors for alcohol relapse.

Variable	Non-Relapse (n = 82)	Relapse (n = 16)	Univariate Odds Ratio (95% CI)	*p*-Value
**Gender**				
Female	3	2	1	
Male	79	14	0.266 (0.041–1.737)	0.167
**Age (y/o) (mean ± SD)**	51.5 ± 7.0	51.6 ± 7.0	1. 002 (0.927–1.082)	0.963
**Donation type**				
Children (son/daughter)	32/26	7/4	1	
Siblings	7	3	2.260 (0.505–10.110)	0.286
Spouse	8	0	0.000 (0.000–2.000)	0.999
Other relatives	6	0	0.000 (0.000–2.000)	0.999
Cadaveric donor	3	2	3.515 (0.525–23.543)	0.195
**Marital status**				
Single	2	1	1	
Married	75	12	0.320 (0.027–3.808)	0.367
Separated/divorced	5	3	1.200 (0.073–19.631)	0.898
**Educational level**				
Primary school	11	5	1	
Junior high school	24	3	0.275 (0.056–1.361)	0.114
Senior high school	38	8	0.463 (0.126–1.705)	0.274
College or above	9	0	0.000 (0.000–0.000)	0.999
**Occupation**				
Housewife/Unemployed	9	2	1	
Retired	63	13	0.929 (0.179–4.808)	0.930
Employed	10	1	0.450 (0.035–5.843)	0.542
**Smoke**				
No	10	1	1	
Yes	72	15	2.083 (0.248–17.523)	0.499
**Diagnosis**				
Alcohol	36	10	1	
Alcohol + HCC	46	6	0.470 (0.156–1.414)	0.179
**Alcohol start age (y/o) (mean ± SD)**	20.6 ± 4.4	17.4 ± 3.1	0.781 (0.647–942)	0.010 *
**Pre-LT abstinence (months)** **(mean ± SD)**	22.1 ± 27.5	13.0 ± 29.6	0.982 (0.951–1.013)	0.257
**Pre-LT abstinence duration**				
<6 months	19	11	1	
>6 months	63	5	0.137 (0.042–4.444)	0.001 *
**HAMD scores (mean ± SD)**	3.5 ± 2.6	4.1 ± 3.5	1.074 (0.895–1.289)	0.443
**APGAR scores**	8.8 ± 1.4	8.6 ± 1.9	0.896 (0.638–1.258)	0.526
**Family history of alcoholism**				
None	41	7	1	
Immediate family	25	5	1.171 (0.335–4.092)	0.804
Siblings	16	4	1.464 (0.377–5.691)	0.582
**Previous diagnosis of mental illness**				
No	73	13	1	
Yes	9	3	1.872 (0.446–7.850)	0.391
**Current diagnosis of mental illness**				
No	69	12	1	
Yes	13	4	1.769 (0.493–6.347)	0.381
**Alcohol use disorders**				
Alcohol abuse	7	1	1	
Alcohol dependence	75	15	1.400 (0.160–12.230)	0.761

Odds ratio and *p*-values are presented; the values within parentheses are 95% confidence intervals. * *p* value < 0.05.

**Table 3 jcm-09-03716-t003:** Multivariate logistic regression analyses to identify independent risk factors for relapse.

Variable	Odds Ratio	95% CI	*p*-Value
Younger age at starting alcohol	1.261	1.039–1.531	0.019
Abstinent <6 months pre-transplant	7.045	2.059–24.106	0.002

Odds ratio and *p*-values are presented; the values within parentheses are 95% confidence intervals.

**Table 4 jcm-09-03716-t004:** Sensitivity, specificity, positive predictive value, and negative predictive value of 6 months abstinence for relapse.

Pre-LT Abstinence Duration	Non-Relapse	Relapse	Total (n)
<6 months	19	11	30
>6 months	63	5	68
	82	16	98
Sensitivity: 0.783	Specificity: 0.688	PPV ^1^: 0.688	NPV ^2^: 0.768

^1^ PPV: positive predictive value; ^2^ NPV: negative predictive value.

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
