# Peer review of "Incidence and Risk Factors of Alcohol Relapse after Liver Transplantation: Analysis of Pre-Transplant Abstinence and Psychosocial Features"

_jcm, 2020, doi:10.3390/jcm9113716_

Round 1
Reviewer 1 Report
Thank you for allowing the revision of this manuscript, which is useful and well-presented. Some changes would improve the manuscript:
- Title: the term prognosis is confusing; the authors may consider use the terms "incidence" and "risk factors" as they did in the introduction. It seems more accurate.
- Abstract: no suggestions
- Introduction: Statement "alcohol relapse after LT is associated with graft loss and worse prognosis" must be detailed (which dimensions: survival? quality of life? graft loss? readmissions?) and counterbalanced with evidence in the opposite sense: https://pubmed.ncbi.nlm.nih.gov/30920118/
- Methods: analyses of baseline characteristics of excluded patients will help to understand better the flowchart. Strength of the methodology is the assessment of smoking status but do the authors have more data? (cigarettes per day, Fagerstöm test,...?). Pretrasplant alcohol abstinence period could be assessed as qualitative variable or quantitative and continuous variable. The second option is more informative. In the method section and results section is not clear. High Risk Alcoholism Relapse (HRAR) predicts relapse after LT in several studies. Do the authors have data about HRAR in their sample? For assessing the alcohol relapse, do the clinical records include biological markers of relapse? Number of relapses identified by self-reporting, relatives reporting and biological markers would be very helpful to understand the reliability of data. Please, provide information about which was the criteria to include the variable in the multivariate analyses.
- Results: the paper would be easier to read if section 3.1 were reduced. MELD average is low. It deserves a comment in the discussion. To understand the relapse rates, the follow-up time duration after LT is required. Multivariate regression analyses showed that age at alcohol starting had less impact on relapse (26.1%) than abstinence < 6 month (704.5%). It deserves a comment in the discussion.
- Discussion: first paragraph is quite repetitive, specially first two sentences. Statement "alcohol relapse seems to be..." is not scientifically appropriate. AUD is an addictive disorder that includes compulsive behaviour and relapses as core symptoms (https://pubmed.ncbi.nlm.nih.gov/27475769/). In addition, "prediction of alcohol relapse" requires evaluation and also cost-effectiveness strategies for relapse prevention. A brief discussion about these strategies would substantially improve the paper (with references). The statement "surveillance" is not appropriate in this context (follow-up or continuous care would be less stigmatising). Regarding family support, authors wrote "96% of the patients underwent LDLT" but in the methods section they wrote that the population was only LDLT. It is a bit confusing. The term reuse is not appropriate (relapse is more appropriate). The statement "psychiatric abstinence group" is not supported in the results. I'd suggest removing it.
- Conclusions: Statement "Alcohol relapse is considered extremely crucial for post-LT compliance and long-term outcomes of LT" and the last sentence of the conclusions are not based on results of the study. Strength of the study is the specific population (LDLT). The paper would improve if the authors decide to include this point in the conclusion.
Reviewer 2 Report
I read the paper from Yu et al with great interest. The authors have looked at the pre-transplant risk factors of alcoholic relapse after liver transplantation for ALD, mainly after living-donor LT, in a single-center retrospective cohort. While their findings are in line with other articles (Egawa et al. Liver Transplantation), there are some original results.
The retrospective setting brings inherent limitations to the paper that make it difficult to determine whether the data obtained is reliable. Most notably the paper is weakened by its retrospective nature and by the absence of clear protocol of detection of alcoholic relapse.
Major concerns:
-The relapse rates are low compared to previous papers and this can be influenced by the duration of the follow-up. Since the cohort is in part recent (less than 5 years since the LT for many patients), the relapse rates have to be interpreted with caution. Moreover, the median follow-up of the cohort should be clarified.
-The definition of alcoholic relapse after LT is difficult ant the one used in this paper is not clear: was "any alcohol consumption" taken into consideration or only "excessive sustained consumtpion"? And if the latter is true, which definition was used?
-The manuscript lacks data on the main outcomes such as patient and graft survival. Indeed, one could argue that relapse taken alone is not per se an outcome after LT for ALD.
-The hospital where the authors work perform a large number of LT and ALD seems like minor indication, however we do not have the total number of LT during the study period, this data would be valuable to appreciate the proportion of LT done for ALD.
Minor concerns:
-Authors should prefer the more recent terminology "Alcohol-associated liver disease" instead of "alcoholic liver disease"
-The title of "Table 1" seems wrong.
-Line 67: authors should clarify what the "early postoperative period" corresponds to.
Round 2
Reviewer 1 Report
Authors have responded properly to the previous comments. Thank you to allowing me the revision of the second version that I really enjoyed.
Just minor comment. If it possible, add a reference to the following statement:
"A possible explanation is that, in comparison with the age of starting alcohol use, the patients with abstinence < 6 months may suggest poor self-control on alcohol use, which enhances the risk of alcohol relapse post-LT"
Reviewer 2 Report
Thank you to the authors for adressing my comments and concerns, as well as those from the other reviewer. The overall quality of the manuscript is increased.
